# Is Site-Specific Pasta a Prospective Asset for a Short Supply Chain?

**DOI:** 10.3390/foods9040477

**Published:** 2020-04-10

**Authors:** Gabriella Pasini, Giovanna Visioli, Francesco Morari

**Affiliations:** 1Department of Agronomy, Food, Natural Resources, Animals and the Environment, University of Padova, Viale dell’ Università 16, 35020 Legnaro-Padua, Italy; gabriella.pasini@unipd.it (G.P.); francesco.morari@unipd.it (F.M.); 2Department of Chemistry, Life Sciences and Environmental Sustainability, University of Parma, Parco Area delle Scienze 11/a, 43124 Parma, Italy

**Keywords:** durum wheat, precision harvest, pasta quality, pasta short chain

## Abstract

In the 2011–2012 season, variable-rate nitrogen (N) fertilization was applied two times during durum wheat vegetative growth in three field areas which differed in soil fertility in northern Italy. The quality traits of the mono-varietal pasta obtained from each management zone were assessed in view of site-specific pasta production for a short supply chain. To this purpose, semolina from cv. *Biensur* obtained from management zones with different fertility treated with N at variable rate was tested in comparison with a commercial reference (cv. *Aureo*) to produce short-cut pasta. *Biensur* semolina demonstrated to have technological characteristics positively correlated with the low-fertility zones treated with high N doses (200 and 200+15 kg/ha) and, to a lesser extent, with the high-soil-fertility zones (130 and 130 + 15 kg/ha of N). The lower quality parameters were obtained for pasta produced with wheat from medium-fertility zones, independently of the N dose applied. The derived pasta obtained from the low-fertility zones treated with high N doses had cooking and sensory properties comparable to those of pasta obtained using the reference cv. *Aureo*. These results are explained by the higher amounts of gluten proteins and by a higher glutenin/gliadin ratio in semolina, which are indicators of technological quality. Overall, the results indicate that segregation of the grain at harvest led to the production of semolina with higher protein content and, hence, to a higher pasta quality. Therefore, site-specific pasta could be a potential asset for a short supply chain, aiming to improve traceability and environmental and economic sustainability.

## 1. Introduction

The key determinants of durum wheat flour quality are the quantity and the quality of gluten proteins. These traits are genetically determined but are also influenced by the climatic conditions and the fertility of a cultivated soil. Northern Italy represents the limit for cultivation of durum wheat in Europe, since in this environment, high-quality standards (i.e., protein content) can be achieved by increasing nitrogen (N) inputs, albeit posing environmental risks [1,2]. In particular, in northern Italy, high rainfall associated with shallow water table conditions and alkaline soils increases the risk of N pollution in water and air [3]. In addition, in uniformly managed fields, high yielding areas or areas with low plant-available soil N may result in low grain protein content due to smaller amounts of available N per kg of grain yield [4].

Recent advances in precision agriculture offer new potentialities to meet grain quality standards. Econometric analyses have shown that the gross income of wheat can be maximized by the combination of variable-rate N fertilization (VR-N) with specific quality criteria [5]. VR-N can in fact reduce the risk of low yields by increasing the probability of maintaining the quality of wheat to the appropriate standard [6].

The amount of protein in grains is an essential parameter for obtaining excellent economic results [7], especially when there are contracts that recognize a quality award. The possibility of carrying out a precision harvest of wheat based on its different quality (i.e., protein content) in the field can be a useful system to achieve the protein requirements established by these contracts [8]. In the two-year experiment on VR-N carried out in a previous study [2], only high N input sandy areas met the standard necessary to be eligible for the premium-quality grain protein content (i.e., 13.5%). Conversely, the average protein content of the entire field was always lower the premium threshold.

The variability of protein content within a field is often huge, justifying the possibility of segregating grains of greater or lesser quality. This procedure can be done at post-harvest in the farm using specific sensors [9,10,11]. Regarding the pre-harvest segregation procedure, it requires the collection of previous information on the quality and yield of cereals [12,13], which could be inferred on the basis of data on management areas for farmers who apply VR-N [14], such as those collected by proximal sensing sensors (e.g., Normalized Difference Vegetation Index). 

Another approach for precision harvesting is to use on-the-go sensors, such as Near-Infrared Reflectance mounted on combine harvesters, which allows the analysis of the percentage of protein during grain flow [9,15]. These sensors could potentially allow to automatically separate wheat grains based on their protein (% N) content [16] and thus are encouraged to be adopted. 

Though different authors have demonstrated that high profits could be achieved by separating different grain qualities during harvest [17], a study evaluating the effect at field scale on pasta quality of spatial variability and the feasibility to produce “site-specific” pasta for a short supply chain is still lacking. The environmental impact of the entire pasta production cycle, from field to packaging, has been reviewed [18]. The authors highlighted that the production of grain (i.e., wheat cultivation) and semolina were the sub-processes that mostly impacted on the environment. Moreover, under a recent Italian decree [19], the origin of the durum wheat grains used in pasta production must be declared. In light of this, an advantageous strategy would be to promote local, short food supply chains in order to improve environmental and economic sustainability [20]. The idea of implementing a short supply chain by adopting selective harvesting strategies was first put in place in the Australian wine market [21]. These authors were able to identify two harvesting zones in a Cabernet Sauvignon vineyard by the use of remote sensing technologies [21]. By segregating grapes in two bins, they produced wines of different quality and, correspondingly, prices. However, a study [22] questioned the scale at which to apply precision harvesting in order to achieve commercial-scale vinification. Bramley and collaborators [23] confirmed the benefits provided by precision harvesting even when grape and wine production is oriented toward large vinification volumes.

In a previous work, we performed an experiment of VR-N on durum wheat in northern Italy with the aim to obtain high yields and protein contents, in relation to field soil properties and N fertilization. According to our previous results, we advanced the hypothesis that protein variability in grain from different field zones could justify the manufacturing of site-specific pasta as a potential asset for a short supply chain. On this basis, in the present study the standard quality parameters of semolina and the corresponding pasta obtained from grain selected in different VR-N management zones were evaluated.

## 2. Materials and Methods

### 2.1. Field Experiment and Grain Production

A field experiment was carried out at the Miana Serraglia farm (NE Italy, 45°22′ N; 12°08′ E) (Mira, Venice, Italy), located close to the Venetian Lagoon, an area classified with a high risk of nitrate leaching in surface waters and ground waters according to the Nitrate Directive 91/676 [24]. The information about field measures and soil texture were previously reported [2]. Durum wheat (*Triticum durum* Desf.) var*. Biensur* (Apsovmenti, Voghera, Italy) was grown in 2010–2011 and 2011–2012. Only grain samples harvested in the second year (seeding 24 October 2011 and harvest 4 July 2012) were analyzed in the present work. Weather was characterized by low rainfall and temperature (1.9 °C compared to the 20-year average of 4.1 °C) at the beginning of stem elongation, which prevented early N uptake.

### 2.2. Selection of Different Management Zones and Variable-Rate Fertilization

A total of 120 samples of the top soil layer (0–30 cm) were collected according to a mixed-sampling scheme [25]. Primary soil properties (texture, bulk density, pH, electric conductivity, organic carbon, total N, and labile phosphorus) were determined according to a previous study [2]. In addition, spatial soil electric conductivity (ECa) was measured with an EMI sensor (Geonics EM38DD). Three management zones (MZs) were delineated, i.e., a high-fertility zone (HFZ), a medium-fertility zone (MFZ), and a low-fertility zone (LFZ). Fertilization with 130, 160, and 200 kg N/ha using ammonium nitrate was applied in HFZ, MFZ, and LFZ, respectively. N doses were defined according to a 30-year model simulation carried out in the three MZs with DSSAT model [26]. Criteria for selecting N doses aimed to balance the productivity with water quality goals (e.g., low N leaching). At the tillering stage, a uniform N rate (52 kg N ha^−1^) was supplied, while VR-N was managed during stem elongation. Due to the high N amount to be distributed, in LFZ, N application was supplied in two applications to avoid N losses. At the flowering stage, each zone was split into a control (0) and a treatment area, and the latter was treated with a UAN (urea–ammonium–nitrate) solution (15 kg N/ha) (Figure 1). Dates and amount of fertilizations for the 2011–2012 season were previously reported [2].

Grain yield was recorded by a yield mapping system (Agrocom CL021) mounted on a combine harvester (Claas Lexion 460). Similarly, protein content was measured with a NIR spectrometer that was interfaced to a GPS, as already reported [2]. Crop production ranged from less than 5 t/ha to more than 7 t/ha (Figure 1), with the low productive areas corresponding to sandier areas. N base fertilization was a key determinant of crop yield, but also foliar N application slightly increased grain yield [2]. Protein content fluctuated from less than 10% to more than 15%, mirroring the VR-N areas (Figure 1). In each of the six combinations of MZs (130; 130 + 15; 160; 160 + 15; 200; 200 + 15) and flowering fertilization, a composite grain sample of 100 kg was collected for the chemical composition and pasta analyses.

### 2.3. Chemical Composition and Gluten Proteins Quantification of Semolina Samples

Grains of the cv. *Biensur* samples (130; 130 + 15; 160; 160 + 15; 200; 200 + 15 kg/ ha) of the 2011–2012 season were ground in an experimental laboratory mill (Buhler MLU202 roller mill; Germany) in order to obtain refined semolina with particle size similar to that of the reference control (200 to 350 µm). The reference control was a commercial semolina (cv. *Aureo*) used for mono-varietal industrial pasta production. The chemical composition of all semolina samples was evaluated according to AOAC standard methods [27] for moisture, protein, starch, fiber, fat, and ash. In addition, the relative quantification of gliadins, high-molecular-weight (HMW) glutenin (GS) fraction, and low-molecular-weight (LMW) glutenin (GS) fraction was carried out using a protein sequential extraction procedure [20] followed by quantification using the Bradford assay [28].

### 2.4. Dough Analyses and Pasta Production

Dough mixing characteristics (water absorption, dough stability, and dough weakening) were measured in triplicate, by using a Promylograph apparatus equipped with a 100 g bowl (T6, Max Egger, Austria) according to the approved 54-21 method [29].

Short-cut pasta (tubetti) (Figure 2) made from both the *Biensur* samples (130; 130 + 15; 160; 160 + 15; 200; 200 + 15) and reference, was produced using an industrial-scale pilot system at the Pavan-Map Impianti (Galliera Veneta, Padova, Italy). Briefly, pasta samples were prepared in accordance with the Italian legislation [30] by mixing water and semolina to form a dough with a 30% moisture content. The dough was processed using a single-screw extruder (FP 70 model, Pavan) under vacuum conditions. The screw speed was 35 rpm, the cylinder and the extrusion head temperature was 35 °C, while the head pressure was 100 bar. Samples of *Biensur* (130; 130 + 15; 160; 160 + 15; 200; 200 + 15) and reference fresh pasta were transferred to the dryer and treated at decreasing air temperatures (from 90 °C to 45 °C) in a static dryer (SD 100 model, Pavan) to obtain the final moisture content of 11%.

### 2.5. Pasta Quality Parameters

Optimal cooking time (OCT), defined as “al dente”, was determined by pressing the pasta between two glass slides at different times during cooking in boiling water and observing the time that the starchy white core of the pasta took to disappear [31].

Cooking loss (CL) was evaluated according to method 66-50 [29]. The residue obtained by draining the pasta cooking water was weighted and reported as a percentage of the starting material.

Water absorption (WA) was calculated as the increase in the weight of the pasta after cooking and expressed as a percentage of the weight of the uncooked pasta.

All analyses were carried out with three individual measurements (replicates).

The firmness of cooked pasta was determined using a Texture Analyser (TA.XT plus, Stable Micro Systems, UK) according to 16-50 method [29]. A single tubetto (12 mm thick) was oriented perpendicularly to a knife probe, then compressed at a speed of 0.5 mm/s with a 5 kg load cell. Firmness was measured as the maximum peak force curve (Newton) required to compress the pasta sample. The average value of five replicates was reported.

### 2.6. Sensory Evaluation of Pasta

To assess the acceptability of the mono-varietal pasta made from cv. *Biensur* obtained from each management zones, a sensory evaluation was carried out by 15 panel members (9 women, 6 men; ages ranging from 22 to 40 years) with experience in food evaluation. The pasta samples (130; 130 + 15; 160; 160 + 15; 200; 200 + 15; reference) were cooked “al dente”, drained, and kept warm until serving in randomized order on plastic plates labelled with random two-digit codes. Panelists were asked to evaluate color, flavor, and texture (firmness and stickiness) on a five-point scale from 1, low intensity, to 5, high intensity. They were also asked to score the overall quality of the product based on these same attributes using the same five-point scale. The attribute scores for each sample and panel member were subjected to a one-way analysis of variance (ANOVA) to obtain mean sensory scores for each of the 15 panel members.

### 2.7. Statistical Analysis

Statistical analysis of the data was performed with the Statgraphics Centurion XIV software (StatPoint Technologies, Inc., Warrenton, VA, USA), and the results were compared with one-way ANOVA. A preliminary Shapiro–Wilk test was applied to test the assumption of normality. Significant differences between treatments were determined by Tukey’s test.

## 3. Results

### 3.1. Chemical Composition of the Semolina and Dough Analyses

Soil variability and VR-N determined the protein content and composition of the semolina from cv. *Biensur* obtained from different management zones (Table 1).

There was a significant higher protein abundance in grain from LFZ (13.6%, on average) and a lower abundance in grain from HFZ (10.4%, on average), independently of the foliar treatment. The protein content in grain from LFZ was similar to that of the commercial high-quality semolina from cv. *Aureo*, taken as a reference sample (14.7%).

Besides the total protein amounts, by sequential gluten protein extraction, the percentages of the different gluten protein classes (gliadins, HMW-GS, and LMW-GS) were also calculated. These are important parameters affecting dough and pasta technological properties. Gluten strength describes the ability of the proteins to form a tenacious network able to promote better extrusion and superior cooking quality and textural characteristics if compared to weak gluten at the same protein level [32,33]. In particular, several studies showed that adding a glutenin-rich fraction consisting of both HMW-GS and LMW-GS to base semolina, increased the mixograph dough strength and the percentage of unextractable polymeric proteins [33].

In addition, it is widely accepted that for the same wheat genotype, the growing environment affects the quality of the semolina. In particular, different works demonstrated that both fertilization dose and fertilization type can modulate the total gluten protein amounts and the relative percentages of different gluten protein fractions (gliadins, HMW-GS, and LMW-GS) both in durum and in common wheat, modifying the HMW-GS/LMW-GS and the glutenin/gliadin (GS/Gli) ratios and consequently the dough strength [34,35,36,37].

In this work, we showed that the percentage of gluten proteins was affected by the different soil fertility zones and the fertilization treatments. In particular, HMW-GS increased from the value for the HFZ (9.5% on average with respect to total gluten proteins) to that for LFZ (12.5%, on average with respect to total gluten proteins), thus suggesting an effect of the N treatment on the synthesis of this class of proteins, independently of soil fertility. LMW-GS showed the lower percentage in semolina from MFZ treated with foliar application (17% average with respect to total gluten proteins) while for both LFZ and HFZ, 23.5% of LMW-GS (on average with respect to total gluten proteins) was observed (Table 2), suggesting that their abundance is related to both the soil fertility and the N supplied. Conversely, the percentage of gliadins appeared to be influenced mainly by the fertility of soil, being more abundant for HFZ and MFZ with respect to LFZ (Table 2). As a result of the different gluten protein proportions, the ratio of total GS/Gli increased for LFZ, indicating a higher quality of the gluten protein composition in these semolina samples. Foliar fertilization influenced the percentage and ratio of the different gluten protein fractions only in grain from MFZ.

We compared all semolina samples for dough stability (length of time the dough maintains its maximum consistency), weakening index (reduction in dough consistency after 20 minutes of mixing), and water absorption (g of water per 100 g of semolina required to reach a dough consistency of 500 PU) (Table 2). The results showed significant differences among cv. *Biensur* samples obtained from different soil fertility zones treated with VR-N (130; 130 + 15; 160; 160 + 15; 200; 200 + 15). In particular, semolina from LFZ, independently from the foliar treatment, showed higher values of dough stability (indicator of the strength) than the other samples, comparable to that of commercial high-quality semolina (>11 min). Samples from LFZ differed also for the amount of water absorbed (ca. 52%) related to protein content (Table 1) and for the dough weakening index (ca. 31), which indicates the tolerance to mechanical mixing. These data indicate that dough properties are also governed by HMW-GS and LMW-GS protein fractions, which are known to be an indicator of strength and coherence of the protein network (Table 1).

### 3.2. Quality Parameters of Pasta

Pasta quality is expressed in terms of water absorption, cooking loss, and firmness (Table 3). The values of water absorption of all pasta samples at the optimal cooking time were similar (Table 3), corresponding to 170.17–1721.38% of weight increase, in line with what found by other authors [38,39], while the cooking loss, that represent the solid substance leaching in water during cooking, was significantly higher for HFZ and MFZ with respect to LFZ, which presented a cooking loss similar to that of the commercial reference (Table 3). Therefore, the results of the LFZ samples indicate that gelatinized starch is well retained by a strong gluten network that is able to form a compact structure, which was also confirmed by the high firmness value (>5 N), measured by Texture Analyzer. The firmness of the cooked pasta resulted positively correlated to the cooking loss (*r* = 0.96), as indicated by statistical analysis. In general, all standard quality parameters did not reveal any significant differences between LFZ pasta, obtained with or without foliar treatment (200 and 200 + 15) of *Biensur*, and *Aureo* pasta, despite *Biensur* having a lower protein content than *Aureo*, as previously confirmed [20], since it had also a good gluten quality, which plays an important role in the formation of a strong protein network.

The sensory properties of the cooked pasta, such as appearance, flavor, and texture (firmness and stickiness), play an essential role in determining the final consumer preference, especially in traditional pasta-consuming countries [40].

Sensory evaluation of pasta (Figure 3) made from the cv. *Biensur* from each management zones and from the reference cv. *Aureo* showed significant (*p* < 0.05) differences between LFZ pasta (with or without foliar treatment, 200 and 200 + 15) and the others, for all the parameters tested (Figure 3). Finally, pasta obtained from LFZ showed good overall acceptability, comparable to that of the reference pasta. As for the consumer’s evaluation on firmness and stickiness, these attributes depend on the intrinsic structure of pasta that is governed by strength and coherence of the gluten network. Therefore, high firmness values correspond to low stickiness values [40,41].

## 4. Conclusions

In the experiments carried out in this work, the variability of grain protein content in different field zones was significant enough to justify the segregation of grain during harvest. In particular, LFZ, associated with a high N content, showed not only the highest total protein amount in the grain, but also the highest technological quality of gluten proteins.

The higher grain quality in the LFZ was reflected in the elevated dough technological characteristics as well as in the pasta cooking behavior and sensory properties, comparable to those of well-established commercial mono-varietal semolina. Although the experiments demonstrated the feasibility to produce “site-specific” pasta, technical constrains could prevent the application of the described strategy at industrial level. Indeed, a minimum grain stock is required in manufacturing pasta, which could not be achieved if the management zones are limited in size. As observed for precision viticulture [21,23], a multi-field scale approach should be followed to guarantee adequate grain stocks. Within a farm, different fields can be considered at the same time, segregating grain within management zones classified according to crop varieties, pedo-climatic conditions, and management (fertilization, seeding, weed control, etc.). In this way, the manufactured “site-specific pasta” could be a potential asset for a short supply chain, aiming to improve traceability and environmental and economic sustainability.

## Figures and Tables

**Figure 1 foods-09-00477-f001:**
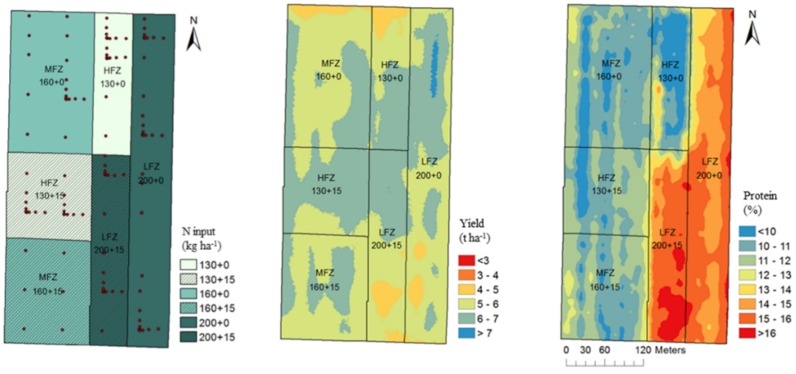
Map of the different fertility zones (**left**), yield (ha; **center**), and protein content (%, **right**) in 2011–2012. In the legend, the N fertilization doses applied were previously reported [2]. HFZ, high-fertility zone, MFZ, medium-fertility zone, LFZ, low-fertility zone. Data derived from reference [2].

**Figure 2 foods-09-00477-f002:**
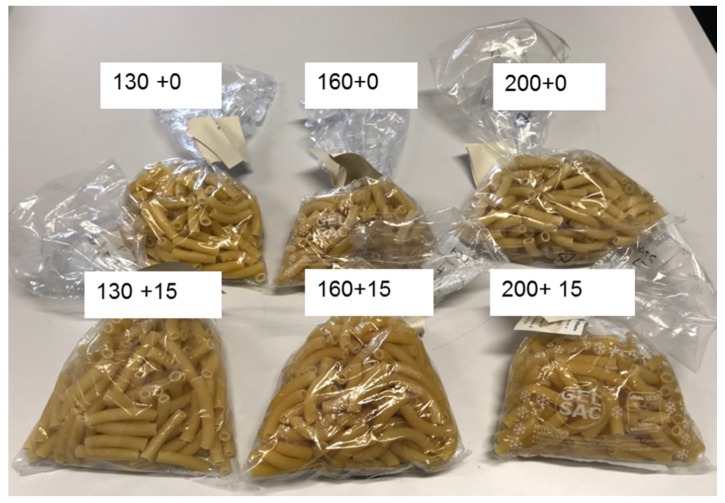
Site-specific pasta manufactured in the management zones.

**Figure 3 foods-09-00477-f003:**
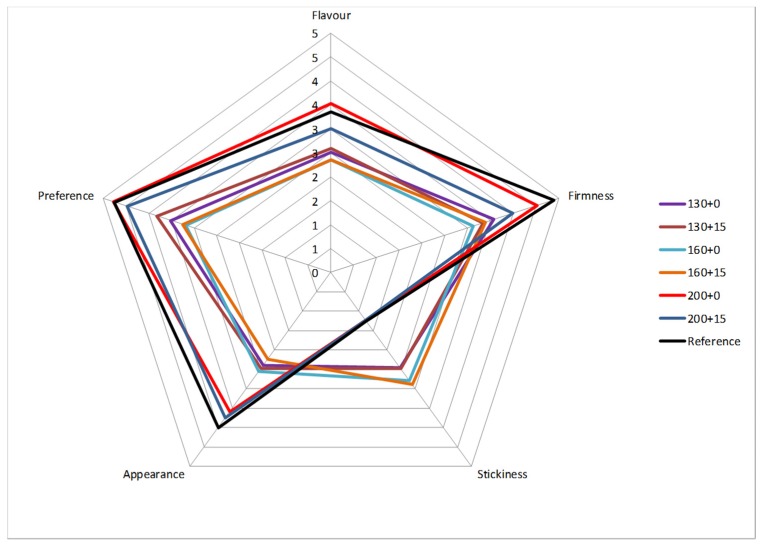
Sensory properties (*n* = 15) of pasta from site-specific harvest of cv. *Biensur* compared with those of the commercial reference cv. *Aureo*. The different fertilization managements listed in the legend are reported in Table 1.

**Table 1 foods-09-00477-t001:** Proximate composition (expressed as g 100/g of dry matter), gluten proteins quantification, and relative percentage of the three classes of gluten proteins (gliadins (Gli), high-molecular-weight glutenins (HMW-GS) and low-molecular-weight glutenins) (LMW-GS) of site-specific semolina samples of cv. *Biensur* compared with the commercial reference cv. *Aureo*. Data are referred to the 2011–2012 cultivation season.

Soil Fertility Zones	Sample Name *	Total Protein ^1^ %	Total Gluten Proteins ^2,^# (mg/g semolina)	% Gli ^3^	% HMW-GS ^3^	% LMW-GS ^3^	HMW/LMW-GS #	Total GS/Gli #	Ash %	Lipids %	Fiber %	Starch %
**HFZ**	130 + 0	10.2 ^c^	16.9 ^b^	71 ^a^	10 ^b^	24 ^b^	0.4 ^b^	0.4 ^b^	0.74 ^a^	1.85 ^a^	3.07 ^a^	82.68 ^b^
**HFZ**	130 + 15	10.6 ^c^	16.8 ^b^	69 ^a^	9 ^b^	22 ^b^	0.4 ^b^	0.4 ^b^	0.77 ^a^	1.90 ^a^	2.97 ^a^	82.18 ^b^
**MFZ**	160 + 0	9.5 ^d^	18.2 ^b^	71 ^a^	9 ^b^	22 ^b^	0.5 ^b^	0.4 ^b^	0.65 ^a^	1.80 ^a^	2.48 ^b^	84.07 ^a^
**MFZ**	160 + 15	9.2 ^d^	17.8 ^b^	70 ^a^	10 ^b^	17 ^c^	0.6 ^a^	0.3 ^b^	0.69 ^a^	1.79 ^a^	2.74 ^b^	85.16 ^a^
**LFZ**	200 + 0	13.9 ^b^	24.2 ^a^	65 ^b^	13 ^a^	26 ^b^	0.5 ^b^	0.6 ^a^	0.75 ^a^	1.81 ^a^	3.04 ^a^	79.78 ^c^
**LFZ**	200 + 15	13.3 ^b^	23.1 ^a^	66 ^b^	12 ^a^	21 ^b^	0.5 ^b^	0.5 ^a^	0.79 ^a^	1.73 ^a^	3.17 ^a^	80.17 ^c^
	Reference	14.7 ^a^	24.2 ^a^	58 ^c^	12 ^a^	30 ^a^	0.4 ^b^	0.7 ^a^	0.78 ^a^	1.80 ^a^	3.01 ^a^	78.86 ^c^

HFZ, high-fertility zone, MFZ, medium-fertility zone, LFZ, low-fertility zone. For each parameter, different letters indicate significant differences (Tukey test, *p* ≤ 0.05; *n* = 5). ^1^ N determined by the Kjeldahl method; ^2^ Total gluten proteins correspond to the sum of the three different protein fractions (gliadins, HMW-GS, and LMW-GS) extracted according to a published procedure [20]. ^3^ Percentage of single classes of proteins related to the value of the sum of the three gluten protein extracts. * The names of the samples correspond to the N fertilization treatment applied (kg/ha) # Data derived from reference [2].

**Table 2 foods-09-00477-t002:** Mixing properties (*n* = 3) of dough samples obtained from site-specific harvest of cv. *Biensur* compared with those of the commercial reference cv. *Aureo*.

Soil Fertility Zones	Sample * Name	Water Absorption (%)	Dough Stability (min)	Dough Weakening (PU)
**HFZ**	130 + 0	49.3 ^c^	7 ^b^	65 ^a^
**HFZ**	130 + 15	49.5 ^c^	7 ^b^	63 ^a^
**MFZ**	160 + 0	48.4 ^c^	6 ^b^	65 ^a^
**MFZ**	160 + 15	48.0 ^c^	6 ^b^	68 ^a^
**LFZ**	200 + 0	51.9 ^b^	11.0 ^a^	32 ^b^
**LFZ**	200 + 15	52.4 ^b^	11.5 ^a^	31 ^b^
	References	55.7 ^a^	12.0 ^a^	25 ^c^

HFZ, high-fertility zone, MFZ, medium-fertility zone, LFZ, low-fertility zone. Within the same column, different letters indicate significant differences (Tukey test, *p* ≤ 0.05); * the names of the samples correspond to the N fertilization treatment applied (kg/ha).

**Table 3 foods-09-00477-t003:** Pasta quality parameters (optimal cooking time (OCT), water absorption, cooking loss (*n* = 3), and firmness) of pasta from site-specific harvest of cv. *Biensur* compared with those of the commercial reference cv. *Aureo.*

Soil Fertility Zones	Sample Name	Cooking Properties			
		OCT (min.sec)	Water Absorption (%)	Cooking Loss (%)	Firmness (N)
**HFZ**	130	8.30	170.46 ^a^	3.53 ^a^	3.83 ^b^
**HFZ**	130 + 15	8.30	170.17 ^a^	3.54 ^a^	3.85 ^b^
**MFZ**	160	8.30	171.17 ^a^	3.94 ^a^	3.68 ^b^
**MFZ**	160 + 15	8.30	170.89 ^a^	3.94 ^a^	3.72 ^b^
**LFZ**	200	9	172.38 ^a^	3.0 ^b^	5.70 ^a^
**LFZ**	200 + 15	9	171.58 ^a^	2.96 ^b^	5.69 ^a^
	Reference	9	171.90 ^a^	2.96 ^b^	5.80 ^a^

HFZ, high-fertility zone, MFZ, medium-fertility zone, LFZ, low-fertility zone. Within the same column, different letters indicate significant differences (Tukey test, *p* ≤ 0.05). * The names of the samples correspond to the N fertilization treatment applied (kg/ha).

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
