# Peer review of "Is Site-Specific Pasta a Prospective Asset for a Short Supply Chain?"

_foods, 2020, doi:10.3390/foods9040477_

Round 1

Reviewer 1 Report

The experimental site can be found on the edge of the Venice Lagoon, which is classified as vulnerable according to the Nitrates Directive 91/676. What does it mean? Are there any restrictions, connected with the amounts of nitrogen applied to the field? Is it possible to apply 200 (200+15) kg of N per ha in such vulnerable areas?

Grain yields are different in 2011 and 2012. This is partially caused by different weather conditions. But what was the preceding crop in 2011? I couldn't find this information in this article, neither in the Morari et al. (2018) - Optimizing durum wheat...Was it legumes, root crops, or other cereals? This is very important information because in one year (2011) you analyze the effect of different fertilizer treatments and the effect of the weather. But next year (2012) the situation has changed rapidly, as you have the same fertilizer treatments, weather (different from 2011, obviously), and different (perhaps) preceding crop. Or maybe it is the same preceding crop as in 2011 - cereal. We, readers, do not know this information. If you have used the same preceding crop, then there is no problem. But if you haven't, it is a problem from the point of view of methodology. This issue ought to be discussed and resolved in your previous paper.

Results:

line 175: "elongation" is missing, and "observed" can be found twice. Typo.

It is not clear from the article from which year the pasta tested by panel members originates. From 2011, or 2012?

Table 2: From which year did these results come from? According to the above-mentioned paper Morari et al. (2018), the total gluten proteins in the LFZ varied from 22.17 mg/g ±0.41 in 2011 to 23.65 mg/g ±0.92
in 2012. But in the current paper, we can see values ranging from 24.2 in the 200+0 fertilizer treatment to 23.1 in the 200+15 treatment.

Grain yield, protein content, the ratio between GS and gliadin, and HMW and LMW-GS concentrations and ratios are the same as the results published in Morari et al. (2018). I understand that the previous paper was focused on the agricultural note of the experiment, analyzing the N balance and efficiency. But similarities between these two papers are enormous. In the current paper, you only added one page of results, analyzing the characteristics of the pasta (water absorption, dough stability...). And 1/2 page focused on the methodology. The final decision is up to the Editor! Honestly, I don't know if this approach is suitable.

What was the initial material for pasta production? I mean the pasta analyzed by panel-members? Grain yield from 2011, as Figure 3 suggests? According to ANOVA, have you tested if data have a normal distribution? This information is missing in the 2.8 part of the manuscript.

Author Response

Answers to Reviewer_1 are listed in bold Italicum

Comments and Suggestions for Authors

The experimental site can be found on the edge of the Venice Lagoon, which is classified as vulnerable according to the Nitrates Directive 91/676. What does it mean? Are there any restrictions, connected with the amounts of nitrogen applied to the field? Is it possible to apply 200 (200+15) kg of N per ha in such vulnerable areas?

Answer: In Vulnerable areas, Nitrates Directive 91/676 caps the manure N input to 170 kg/ha/y-1, averaged over the farm area. Actually, in our experiment we used only mineral fertilizer. Most recently, the Veneto Regional Government limited also the mineral N input, which in case of durum wheat corresponds to a maximum 190 kg/ha/-y. Even considering this recent directive, the average N-VR input in our experiment meets with regional requirements.

Grain yields are different in 2011 and 2012. This is partially caused by different weather conditions. But what was the preceding crop in 2011? I couldn't find this information in this article, neither in the Morari et al. (2018) - Optimizing durum wheat. Was it legumes, root crops, or other cereals? This is very important information because in one year (2011) you analyze the effect of different fertilizer treatments and the effect of the weather. But next year (2012) the situation has changed rapidly, as you have the same fertilizer treatments, weather (different from 2011, obviously), and different (perhaps) preceding crop. Or maybe it is the same preceding crop as in 2011 - cereal. We, readers, do not know this information. If you have used the same preceding crop, then there is no problem. But if you haven't, it is a problem from the point of view of methodology. This issue ought to be discussed and resolved in your previous paper.

Answer: We thanks the reviewer for the comments. We have better specified the crop succession in the new MS. Data analysed in this paper are those collected in the year 2011-2012. The preceding crop (2010-2011) was durum wheat var. Biensur, as in the second year, fertilized according to an identical N-VR fertilisation plan.

Results:

line 175: "elongation" is missing, and "observed" can be found twice. Typo.

Answer: DONE

It is not clear from the article from which year the pasta tested by panel members originates. From 2011, or 2012?

Answer: the pasta tested by panel members was produced in 2012. The information was added in the Materials and Methods section

Table 2: From which year did these results come from? According to the above-mentioned paper Morari et al. (2018), the total gluten proteins in the LFZ varied from 22.17 mg/g ±0.41 in 2011 to 23.65 mg/g ±0.92
in 2012. But in the current paper, we can see values ranging from 24.2 in the 200+0 fertilizer treatment to 23.1 in the 200+15 treatment.

Answer: We can understand the query of the referee since we made a mistake. In fact, we wrongly inserted the yield and protein maps of year 2010-2011 season. The pasta production was performed collecting grains in June 2012 (season 2011-2012). We clarified this aspect in the different parts of the paper. Table 2 now “Table 1” reports data of “Season 2011-2012” as explained and the maps were changed accordingly.  

Grain yield, protein content, the ratio between GS and gliadin, and HMW and LMW-GS concentrations and ratios are the same as the results published in Morari et al. (2018). I understand that the previous paper was focused on the agricultural note of the experiment, analyzing the N balance and efficiency. But similarities between these two papers are enormous. In the current paper, you only added one page of results, analyzing the characteristics of the pasta (water absorption, dough stability...). And 1/2 page focused on the methodology. The final decision is up to the Editor! Honestly, I don't know if this approach is suitable.

Answer: For the sake of clarity, we would like to strongly underline that methods and data already published in Morari et al, 2018 were reported in this MS only because they are instrumental to interprete results on dough and Pasta characteristics. In the old MS we “honestly” reported their source making reference to the previous work.

Anyway, we clearly understand the reviewer’s concern. Accordingly, we modified the new MS as follows:

  • Published results on yield and protein content (maps reported in old Fig. 3) were moved in the Method section 2.2 and Fig. 1.
  • As far as possible, Methods section were shortened making reference to the previous paper (e.g. old Table 1)
  • Data of the previous paper in “Table 2” were left in the new “Table 1”, labelling which columns contain data derived from the previous paper.

What was the initial material for pasta production? I mean the pasta analyzed by panel-members? Grain yield from 2011, as Figure 3 suggests? According to ANOVA, have you tested if data have a normal distribution? This information is missing in the 2.8 part of the manuscript.

Answer: the pasta tested by panel members was produced in 2012. Moreover, Shapiro-Wilk test was applied to test the assumption of normality. The information has been added in the text (part 2.7)

Reviewer 2 Report

I believe that the paper Is site-specific pasta a prospective asset for a short supply chain in the northern Mediterranean environment? it is an interesting one.  The article presents a lot of data but is not very focus on the wheat flour characteristics on pasta quality. The article presents in a quite extensive way the agronomy part of grains used as raw material for wheat flour production. The authors make even the analysis for the gluten structure but they do not explain very well if this structure is proper for a good wheat pasta quality. In my opinion a connection part of the article is misssing. The authors focus on grains, pasta quality but not too much on wheat flour quality for pasta. So more explanations about how wheat flour quality for pasta should be according to their data reported must be done.

Other observations:

Introduction part: line 34 – the quantity and the composition of gluten proteins must be change with the quantity and quality of gluten proteins

Materials and methods: 2.4 Dough Farinographic analysis. Accodring to the authors the device used is not a Farinograph is a Promylograph. So please revise in entire manuscript the fact that a Promylograph is use and not a Farinograph. They offer similar results but is not the same device.  Also, in my opinion the Promylograph indexes are not correctly put by the auhors to the characteristics of the pasta section (the data should be placed separately to the dough pasta characteristics)

Page 4 lines 133-135, the authors declare  The dough was driven through a vacuum system then extruded to  mould the pasta... is not very clear for me how pasta was made, in what extruded device (model, type), how the pasta was dry, e.g. Please explain in a more detailed way the technological process use to obtain the pasta.

Page 6 lines 200-201 the authors declare that HMW-GS and LMW-GS are known to be directly related to dough strength and extensibility – is not very clear to me which influence on who. Please explain.

Page 1 of 12 lines 235-237. The authors declare that Good quality pasta should meet the criteria of high water absorption, low cooking losses and good texture [40, 41], showing firm  enough to resist surface disintegration and have no excessive adhesiveness but in my opinion the data shows low water absortion. The authors must offer criteria for WA, dough stability and dough weakening, if their parameters met the criteria requested for dough pasta.

Author Response

Answers to Reviewer_2 are listed in bold Italicum

Comments and Suggestions for Authors

I believe that the paper Is site-specific pasta a prospective asset for a short supply chain in the northern Mediterranean environment? it is an interesting one.  The article presents a lot of data but is not very focus on the wheat flour characteristics on pasta quality. The article presents in a quite extensive way the agronomy part of grains used as raw material for wheat flour production. The authors make even the analysis for the gluten structure but they do not explain very well if this structure is proper for a good wheat pasta quality. In my opinion a connection part of the article is misssing. The authors focus on grains, pasta quality but not too much on wheat flour quality for pasta. So more explanations about how wheat flour quality for pasta should be according to their data reported must be done.

Answer: We thank the reviewer for the interest in this work. In this revised version of the manuscript we considered his/her suggestions. In particular, we implemented the introduction section explaining better the novelty of this work by making examples that consider the application of Precision Agriculture and Precision Harvest to other cultivars (Lines 74-84). In addition, according to reviewer suggestions we tried to stress throughout the paper which data are more important to evaluate the flour characteristics and pasta quality. As example, we reported some previous works underlining the importance of Glutenins in determining dough strength and stability (Lines 233-266).

Other observations:

Introduction part: line 34 – the quantity and the composition of gluten proteins must be change with the quantity and quality of gluten proteins

Answer: DONE

Materials and methods: 2.4 Dough Farinographic analysis. Accodring to the authors the device used is not a Farinograph is a Promylograph. So please revise in entire manuscript the fact that a Promylograph is use and not a Farinograph. They offer similar results but is not the same device.  Also, in my opinion the Promylograph indexes are not correctly put by the auhors to the characteristics of the pasta section (the data should be placed separately to the dough pasta characteristics)

Answer: the reviewer is right. Egger Promylograph is a device for determination of flour quality in relation to mixing, working similarly to Brabender farinograph. The text has been correct. Moreover, the promylograph indexes (Table 2) has been moved to paragraph 3.1

Page 4 lines 133-135, the authors declare The dough was driven through a vacuum system then extruded to mould the pasta... is not very clear for e how pasta was made, in what extruded device (model, type), how the pasta was dry, e.g. Please explain in a more detailed way the technological process use to obtain the pasta.

Answer: the reviewer is right. More details on technological process have been added in the text.

Page 6 lines 200-201 the authors declare that HMW-GS and LMW-GS are known to be directly related to dough strength and extensibility – is not very clear to me which influence on who. Please explain.

Answer: We added some information regarding this point at Lines 233-241…. Besides the total protein amounts, by sequential gluten protein extraction, the percentages of the different gluten protein classes (gliadins, HMW-GS and LMW-GS) are also calculated. These are important parameters affecting dough and pasta technological properties. Gluten strength describe the ability of the proteins to form a tenacious network able to promotes better extrusion properties and superior cooking quality and textural characteristics if compared to weak gluten of the same protein level [33 and references within]. In particular, several studies showed that, adding a glutenin rich fraction consisting in both HMW and LMW-GS to base semolina, there was an increase in the mixograph dough strength and in the percentage of unextractable polymeric proteins [33 and references within]”.

Page 1 of 12 lines 235-237. The authors declare that Good quality pasta should meet the criteria of high water absorption, low cooking losses and good texture [40, 41], showing firm enough to resist surface disintegration and have no excessive adhesiveness but in my opinion the data shows low water absortion.

The authors must offer criteria for WA, dough stability and dough weakening, if their parameters met the criteria requested for dough pasta.

Answer: The reviewer is right. The text has been changed. The authors intend to refer to the reference control (semolina and pasta) and not to absolute criteria. The reference is a high quality commercial semolina (cv. Aureo) already used for mono-varietal industrial pasta production. Moreover, the comment “in my opinion the data shows low water absorption" is correct because of a misprint. The error has been corrected.

Round 2

Reviewer 1 Report

The reviewed paper is well corrected.

Reviewer 2 Report

I believe that the authors mad a high effort to correct this manuscript. They took my opinion into account and they correct all I requested.